# Support Vector Machine Methods and Artificial Neural Networks Used for the Development of Bankruptcy Prediction Models and their Comparison

**Jakub Horak *** , **Jaromir Vrbka** and **Petr Suler**

School of Expertness and Valuation, Institute of Technology and Business in Ceske Budejovice, Okruzni 517/10, 37001 Ceske Budejovice, Czech Republic; vrbka@mail.vstecb.cz (J.V.); petr.suler@cez.cz (P.S.)
* Correspondence: horak@mail.vstecb.cz

**Abstract:** Bankruptcy prediction is always a topical issue. The activities of all business entities are directly or indirectly affected by various external and internal factors that may influence a company in insolvency and lead to bankruptcy. It is important to find a suitable tool to assess the future development of any company in the market. The objective of this paper is to create a model for predicting potential bankruptcy of companies using suitable classification methods, namely Support Vector Machine and artificial neural networks, and to evaluate the results of the methods used. The data (balance sheets and profit and loss accounts) of industrial companies operating in the Czech Republic for the last 5 marketing years were used. For the application of classification methods, TIBCO's Statistica software, version 13, is used. In total, 6 models were created and subsequently compared with each other, while the most successful one applicable in practice is the model determined by the neural structure 2.MLP 22-9-2. The model of Support Vector Machine shows a relatively high accuracy, but it is not applicable in the structure of correct classifications.

**Keywords:** neural networks; support vector machine; bankruptcy model; prediction; bankruptcy

---

## 1. Introduction

In financial bankruptcy analysis, the diagnosis of companies at risk for bankruptcy is crucial in preparing to hedge against any financial damage the at-risk firms stand to inflict (Kim et al. 2018). According to Rybárová et al. (2016), bankruptcy models are early warning systems based on the analysis of selected indicators able to identify a thread for financial health of a company. Kiaupaite-Grushniene (2016) states that creation of reliable models of bankruptcy prediction is essential for various decision-making processes. According to Mousavi et al. (2015), frequently used models are mainly Altman Z-Score, Taffler Z-Score, and Index IN95.

A wide number of academic researchers from all over the world have been developing corporate bankruptcy prediction models, based on various modelling techniques. Numerous statistical methods have been developed (Balcaen and Ooghe 2004). Despite the popularity of the classic statistical methods, significant problems relating to the application of these methods to corporate bankruptcy prediction remain. Problems related to statistical methods according to Balcaen and Ooghe (2004, p. 1):

1.  The dichotomous dependent variable,
2.  The sampling method,
3.  Nonstationarity and data instability,
4.  The use of annual account information,
5.  The selection of the independent variables,

6.    The time dimension.

For the purpose of this article, Support Vector Machines (SVM) and artificial neural networks are used. These two methods have been used by many authors to predict corporate bankruptcy, and their results suggest that these two methods are more appropriate than traditional statistical methods (Shin et al. 2005; Xu et al. 2006; Kim et al. 2018; Vochozka and Machová 2018; Machová and Vochozka 2019; Krulický 2019). SVM is sensitive to model form, parameter setting and features selection. SVM, firstly developed by Vapnik in 1995 (Vapnik 1995), is a supervised learning model with associated learning algorithms that analyze data and recognize patterns, used for classification and regression analysis (Burges 1998). According to Lu et al. (2015), compared with other algorithms, SVM has many unique advantages when applied in solving small sample, nonlinear, and high-dimensional pattern recognition problem. The concept of a neural network has been developed in biology and psychology, but its use goes to other areas, such as business and economics (Vochozka 2017). They are especially valuable where inputs are highly correlated, missing, or there are nonlinear systems and they can capture relatively complex phenomena (Enke and Thaworwong 2005). Like any method, SVM or artificial neural networks have disadvantages. Although SVM or artificial neural networks have a good performance on classification accuracy, one main disadvantage of these methods is the difficulty in interpreting the results (Härdle et al. 2009).

The aim of the paper is to develop bankruptcy prediction models and compare results of different methods using classification methods, namely Support Vector Machines and artificial neural networks (multilayer perceptron artificial neural networks—MLP and radial basis function artificial neural networks—RBF). Further to the defined goal, we will ask a research question: "Are artificial neural networks (also NN) more accurate in predicting bankruptcy than SVM?"

The article meets the formal criteria of a scientific text. In the part of literature review there are described methods for evaluation of corporate bankruptcy, attention is paid to artificial neural networks and SVM methods. The methodological part describes used data for the calculation, specifies the particular variables used and presents two above mentioned methods. In the results part there are presented the results achieved by SVM method, then the results obtained by artificial neural networks and the results of both methods are compared. The results are also compared with the results of other authors and the added value of the article is defined. The final part summarizes the results, presents the variables that have the greatest predictive power and suggests further research in this area.

## 2. Literature Review

Company activities are directly or indirectly influenced by various external and internal factors (Boguslauskas and Adlyte 2010). Purvinis et al. (2005) argue that unfavourable business environment, risky decisions of business managers, and unexpected and disadvantageous events may influence a company in insolvency and lead to bankruptcy. Hafiz et al. (2015) state that bankruptcy models are mainly needed by financial entities, e.g., banks. Their advantage consists especially in their ability to provide clear information about potential risks and eliminate such problems in a timely manner. They are important for current and future decision-making (López Iturriaga and Sanz 2015). Predictive models of financial bankruptcy enable to take timely strategic measures in order to avoid financial distress (Baran 2007). For other stakeholders, such as banks, effective and automated rating tools will enable to identify possible financial distress of potential clients (Gestel et al. 2006). The ability to accurately predict business failure is a very important topic in financial decision-making (Mulačová 2012).

A very useful tool to predict the development of companies going to bankrupt is by using artificial neural networks (ANNs) or Support Vector Machine (SVM). Currently, neural networks are applicable in various areas. ANNS are used for solving possible future difficulties, e.g., for predicting company bankruptcy (Pao 2008; Klieštik 2013). Sayadi et al. (2014) state that their main advantages are the ability to generalize and to learn. According to Machová and Vochozka (2019), the disadvantages of ANNs include possible illogical behaviour of networks and required high quality data. Vochozka and Machová (2018) state that ANNs are currently one of the most popular prediction methods.

The SVM method has become a powerful tool for solving problems in machine learning. Many SVM algorithms include solving of convex problems, such as linear programming, quadratic programming, as well as nonconvex and more general problems with optimization, such as integer programming, bilevel programming, etc. However, there are also certain disadvantages of SVM. An important issue that has not been solved fully is choosing the parameters of the core functions. In practical terms, the crucial problem of SVM is its high algorithmic complexity and extensive requirements for the memory of required quadratic programming in complex tasks (Tian et al. 2012).

The aim of Erdogan (2013) was to apply the SVM method in analysing bank bankruptcy. In this work, the SVM method was applied for the analysis of financial indicators. The author states that SVB is able to extract useful information from financial data and can thus be used as a part of early warning system. Chen and Chen (2011) state that the prediction of financial crisis of a company is an important and widely discussed topic. They used particle swarm optimization (PSO) to obtain optimized parameter settings for the SVM method. Moreover, they used the PSO's integrated commitment with the SVM approach to create a model of predicting financial crisis. Experimental results have shown that the approach is efficient in finding better parameter settings and significantly improves the success rate in predicting company financial crisis. Since financial indicators are independent variables, Park and Hancer (2012) applied ANNs on bankruptcy of a company operating in catering and compared the results with the results of logit model. On the basis of empirical results of these two methodologies, ANNs showed higher accuracy than logit model in sample testing. Dorneanu et al. (2011) use ANNs for predicting company bankruptcy. According to the authors, the use of ANNs for the prediction is extremely effective, since the percentage of prediction accuracy is higher than in the case of using conventional methods. The objective of Kim (2011) is to provide an optimal approach to company bankruptcy predicting and to explore functional characteristics of multivariate discriminant analysis, ANNs and the SVM method in predicting the bankruptcy of a specific company. The results have shown that ANNs and SVM are models applicable for predicting company bankruptcy and show promising results. On the basis of the information obtained, the objective of this paper can be considered relevant.

## 3. Materials and Methods

The Albertina database will be the source of data concerning industrial companies operating in the Czech Republic. In terms of sufficient amount of data and in particular the number of companies in liquidation and thus the relevance of the results, more fields within section C—Manufacturing of the CZ-NACE (comes from French – Czech Nomenclature statistique des Activités économiques dans la Communauté Européenne) = Classification of Economic Activities, will be used, namely in the groups 10–33:

- 10: Manufacture of food products.
- 11: Manufacture of beverages.
- 12: Manufacture of tobacco products.
- 13: Manufacture of textiles.
- 14: Manufacture of wearing apparel.
- 15: Manufacture of leather and related products.
- 16: Manufacture of wood and products of wood and cork, except furniture.
- 17: Manufacture of paper and paper products.
- 18: Printing and reproduction of recorded media.
- 19: Manufacture of coke and refined petroleum products.
- 20: Manufacture of chemicals and chemical products.
- 21: Manufacture of basic pharmaceutical products and pharmaceutical preparations.
- 22: Manufacture of rubber and plastic products.
- 23: Manufacture of other non-metallic mineral products.
- 24: Manufacture of basic metals; foundry.

- 25: Manufacture of fabricated metal products, except machinery and equipment.
- 26: Manufacture of computer, electronic and optical products.
- 27: Manufacture of electrical equipment.
- 28: Manufacture of machinery and equipment.
- 29: Manufacture of motor vehicles (except motorcycles), trailers and semi-trailers.
- 30: Manufacture of other transport equipment.
- 31: Manufacture of furniture.
- 32: Other manufacturing.
- 33: Repairs and installation of machinery and equipment.

For the same reasons, the selection of data will not be limited by the size of companies and the number of employees. The output will thus be applicable not only in specific companies, but basically in the whole economic sector.

The data series will consist of five consecutive fiscal years—for each year all the companies in liquidation will be selected and similarly, randomly selected three times the number of active enterprises. The numbers of companies for individual years are then as follows:

- Year 2013: 488 in liquidation, 1464 active,
- Year 2014: 416 in liquidation, 1248 active,
- Year 2015: 354 in liquidation, 1062 active,
- Year 2016: 287 in liquidation, 862 active,
- Year 2017: 163 in liquidation, 489 active.

The same companies will be selected for each year. Different numbers are due to the fact that some companies went bankrupt during the monitored period, ceased to be active and went into liquidation, etc. The sample starts in 2013, that is, in the period of constant economic growth following the period of economic crisis. The authors tried to avoid the results of the models to be affected by economic crisis.

Financial statements, specifically balance sheets and profit and loss statements of all the above mentioned companies will be analysed. Table 1 shows selected financial data and their averages per individual years.

**Table 1.** Selected financial data of data sample.

| Active Companies | | | | | | |
|---|---|---|---|---|---|---|
| **Financial Data** | **2013** | **2014** | **2015** | **2016** | **2017** | **Total** |
| Total assets | 113,590.43 | 112,398.89 | 72,359.06 | 92,463.05 | 102,843.14 | 91,228.91 |
| Fixed assets | 51,794.64 | 48,418.16 | 32,899.65 | 41,244.60 | 49,662.11 | 40,808.34 |
| Current assets | 61,093.99 | 63,352.86 | 38,750.26 | 50,275.42 | 52,550.73 | 49,762.88 |
| Liabilities in total | 113,590.43 | 112,398.89 | 72,359.06 | 92,358.40 | 102,843.14 | 91,228.91 |
| Equity | 51,663.05 | 53,660.95 | 39,599.74 | 42,971.99 | 59,471.72 | 44,077.03 |
| Borrowed capital | 61,076.00 | 58,079.58 | 32,437.83 | 46,616.22 | 42,632.01 | 46,275.94 |
| Operating result | 1574.10 | 14,159.14 | 4604.23 | 7104.33 | 10,576.95 | 6263.25 |
| Economic result for accounting period | 1282.11 | 11,387.14 | 3168.16 | 5231.82 | 9325.24 | 4916.57 |
| Companies in Liquidation | | | | | | |
| **Financial Data** | **2013** | **2014** | **2015** | **2016** | **2017** | **Total** |
| Total assets | 22,033.59 | 21,401.33 | 20,401.53 | 14,201.20 | 10,273.09 | 77,297.73 |
| Fixed assets | 6307.66 | 6768.54 | 5231.99 | 5439.59 | 1481.12 | 33,904.17 |
| Current assets | 15,615.94 | 14,447.03 | 15,116.56 | 8639.67 | 8670.90 | 42,801.79 |
| Liabilities in total | 22,033.23 | 21,390.77 | 20,400.42 | 14,201.20 | 10,273.09 | 77,254.27 |
| Equity | 5454.03 | 5998.58 | 7499.03 | 1768.26 | 2140.76 | 37,917.64 |
| Borrowed capital | 16,453.56 | 15,338.02 | 12,813.47 | 12,382.05 | 8064.87 | 38,566.51 |
| Operating result | −1791.99 | −166.49 | −1219.85 | 214.53 | 284.29 | 7107.70 |
| Economic result for accounting period | −1910.42 | −151.22 | −1492.21 | 116.79 | 141.06 | 5516.93 |

Note: all data in the Table are given in thousands of CZK. Source: own construction.

The data will be checked. Only the data that, at first sight, is not defective or intentionally distorted will be kept on the file for further analysis. This will eliminate record lines (a line represents financial statements per company and year) including:

1. Different assets and liabilities balance,
2. Negative assets,
3. Negative fixed assets,
4. Negative tangible fixed assets,
5. Negative current assets,
6. Negative financial assets,
7. Negative inventories.

The input continuous variables will be:

- AKTIVACELK—Total assets resulting from past economic operations. Thus it means the future economic benefit of the company.
- STALAA—Fixed assets are long-term, fixed and noncurrent. This item includes asset components used for the company business in a long term (more than 1 year) and consumed over time.
- HIM—Intangible fixed assets will depreciate, expressed by the level of depreciation. Intangible fixed assets have a significant impact on the value of the enterprise, they maintain their value for a longer time and are not exposed to the fast operating cycle.
- OBEZNAA—Current assets characterize the operating cycle. They continuously circulate and change their form. They include cash, material, semi-finished products, work in progress, products, or receivables from customers.
- Z—Inventories are current (short-term) assets of the company. They are consumed during operation. In general, inventories include material, inventories for production of its own products and goods
- KP—Short-term receivables are payable in less than 1 year from the date when their arise and represent the creditor's right to seek fulfilment of a certain obligation from the other party, the receivable is extinguished when the obligation is paid.
- FM—Financial assets including long-term and short-term financial assets. Long-term financial assets hold their value for a longer period of time, they do not change into cash quickly. They include securities, bonds, certificates of deposit, obligations, term deposits or loans granted to companies. Short-term financial assets are used for operation, especially for payment of liabilities. Short-term assets represent high liquidity; the expected holding is less than one year. They mainly include money in bank accounts, treasury, checks, clearing notes, valuables or short-term securities and shares.
- PASIVACELK—Total liabilities—information concerning the source to cover the company's assets.
- VLASTNIJM—Equity is the internal source of finance for business assets and capital formation. It includes, in particular, contributions of the founders (owners or partners) to the capital stock and components arising from the business management.
- FTZZ—Reserve funds, undistributable reserves and other funds from profit represent the company's internal sources of finance increasing the company's equity without changing its capital stock. Reserve funds are used as internal resources to cover future losses of the company. Undistributable reserves are created by cooperatives also to cover the loss.
- HVML—Profit/loss brought forward is part of liabilities, an item of equity. These are resources created after tax in previous years. These are funds which are not transferred to funds or distributed and paid. It consists of three parts - retained earnings, loss carried forward and other profit/loss brought forward.

- HVUO—Profit and loss of the current financial period is the sum of profit and loss from operations and financial activities in the financial period and the profit before tax. For calculation, the income tax for ordinary activities is deducted.
- CIZIZDROJE—External resources are the company's debts which must be paid within a certain period of time. These are the company's payables to other entities.
- KZ—Current liabilities are payable within 1 year and used for financing (together with equity) of the normal operation of the company. In particular, they include short-term bank loans, payables to employees and institutions, debts to suppliers or delinquent tax.
- V—Production is goods and services that are used to meet the needs. They result from business activities of the company and characterize the main business activities—production.
- VS—Production consumption mainly includes the costs of consumed material, energy, travel expenses, maintenance and repairs, or low-value assets. It is a sum item which correlates with consumption of materials, services and energy.
- SPMAAEN—Material and energy consumption is an item accounting for inventories - current assets. Energy consumption rises proportionally and positively correlates with the production volume. However, material costs may decrease as the production volume increases. Material consumption is directly dependent on consumption standards and purchase prices.
- SLUZBY—Services are systematic external activities that satisfy human needs, or the business needs in their own course.
- PRIDHODN—Value added represents the sales margin, sales, stock level changes of internally produced inventories, or capitalization less production consumption. It includes the sales margin as well as production.
- MZDN—Payroll costs generally comprise of the employee's gross wages and premiums paid by the employer for each employee's social security and health insurance.
- NNSOCZAB—Employee's social security and health insurance costs.
- OHANIM—Depreciation of intangible and tangible fixed assets provides a tool for gradually assigning the value of fixed assets to expenses. Therefore, it means a gradual assignment of the fixed asset cost value to expenses. It represents depreciation of fixed assets.

The categorical output variable will be considered as:

- STAV—Identifies the situation of the company whether active or in liquidation. There will only be two possible outcomes.

The variables were chosen so that it was possible to express the main features of the company´s capital structure, sources of assets financing, corporate payment history, customers´ payment history, cost structure, and the ability to generate outcomes (sales) and realized added value. The selection of indicators is based on the analysis of the existing Altman Z-Score (Altman 1968, 2000, 2003; Altman and Hotchkiss 2006), IN (Neumaierová and Neumaier 2005, 2008), Taffler index (Taffler and Tisshaw 1977; Taffler 1983), Kralicek Quick Test (Kralicek 1993), Harry Pollak´s method (Pollak 2003), and Vochozka´s method (Vochozka 2010; Vochozka and Sheng 2016; Vochozka et al. 2017). The conditions of external environment are not considered, as all companies in the dataset operate in one market, and therefore they are all influenced equally. The output is thus analogy to certain extent. If patterns are identified (although given by a large number of input variables combinations), it is possible to observe a similar development of two companies showing just about the same combination of input variables on the basis of similarity.

The Statistica software, version 13 of TIBCO will be used to apply the classification methods.

### 3.1. Support Vector Machines

Machine Learning option in the Data Mining module will be used to apply SVM. The file will be divided into a train (75%) and a test (25%) data subset. Then SVM type 2 will be specified where the error function is identified as:

$$\frac{1}{2}w^T w - C\left[v\varepsilon + \frac{1}{N}\sum_{i=1}^{N}(\zeta_i + \zeta_{i\cdot})\right],$$ (1)

which minimizes the entity to:

$$\begin{aligned}
\left[w^T \varnothing(x_1) + b\right] - y_i &\leq \varepsilon + \zeta_i \\
y_i - \left[w^T \varnothing(x_1) + b_i\right] &\leq \varepsilon + \zeta_{\cdot i} \\
\zeta_i, \ \zeta_{\cdot i} &\geq 0, \ i = 1, \ \ldots, \ N, \ \varepsilon \geq 0.
\end{aligned}$$ (2)

Then the SVM (kernel function) will be selected. In this case, it will be Sigmoid that should be able to identify the extreme values:

$$K(X_i, X_j) = \tan h(\gamma X_i \cdot X_j + C),$$ (3)

where $K(X_i, X_j) = \varphi(X_i) \cdot \varphi(X_j)$, which means that SVM function represents an output value of input variables projected in multidimensional space using transformation $\varphi$.

The results (value 10, seed 1000) will then be cross-validated. A maximum of 10,000 iterations will be performed with a possible ending in case of the error 0.000001.

### 3.2. Artificial Neural Networks

Classification analysis based on multilayer perceptron neural networks and radial basis function neural networks. ANS (automatic neural network) mode will be used. In case of unsatisfactory results, the result may be corrected using the custom network designer.

The set will be divided by random into three groups of enterprises—i.e., a train file (where neural networks are trained to achieve the best results)—70% of the data, a test file (identify if the classification of trained neural structures is successful)—15% data and a validation file (used for additional verification of the result)—15% of data. Only MLP and RBF will be used in the calculation. For MLP networks, the minimum number of hidden neurons will be set to 8 and the maximum number to 25 while for RBF, the minimum will be 21 and the maximum will be 30 hidden neurons. The number of networks for training will be 10,000 whereas 5 networks with the best results will be retained. The error function will be the sum of squares:

$$E_{SOS} = \frac{1}{2N}\sum_{i=1}^{N}(y_i - t_i)^2,$$ (4)

where $N$ is number of training cases, $y_i$ is predicted target variable $t_i$, $t_i$ is target variable of a $i$-th case.

The BFGS algorithm (Broyden–Fletcher–Goldfrarb–Shanno) will be used for calculation, for more details see Bishop (1995).

Another error function will be entropy (or, cross entropy error function):

$$E_{CE} = \sum_{i=1}^{N} t_i \ln\left(\frac{y_i}{t_i}\right),$$ (5)

The activation functions shown in Table 2 will be considered for NN.

**Table 2.** Activation functions of MLP and RBF hidden and output layer.

| Function | Definition | Range |
|---|---|---|
| Identity | $a$ | $(-\infty; +\infty)$ |
| Logistic sigmoid | $\frac{1}{1+e^{-a}}$ | $(0;1)$ |
| Hyperbolic tangent | $\frac{e^a-e^{-a}}{e^a+e^{-a}}$ | $(-1;+1)$ |
| Exponential | $e^{-a}$ | $(0; +\infty)$ |
| Sine | $\sin(a)$ | $[0;1]$ |
| Softmax | $\frac{\exp(a_i)}{\sum \exp(a_i)}$ | $[0;1]$ |
| Gaussian | $\frac{1}{\sqrt{2\pi\sigma}} \exp\left[-\frac{(x-\mu)^2}{2\sigma^2}\right]$ | |

Source: own construction.

Neural networks work as follows: the data of a specific company are entered and subsequently, as an independent variable, the data are converted using the activation function and weights into the values of hidden neurons, which are the input variables for the second round of calculation. Here, the activation function and trained weights as used as well. The result obtained is subsequently compared at a given interval, and it is determined whether or not the company is able to survive possible financial distress.

Other settings will remain default. The result will be a bankruptcy model (the development of the company will be evaluated using two variables—survival of the company or a bankruptcy tendency—thus, the dependent variable will only take two values 0 or 1). The model development will be an iterative and recurrent process with actions to improve. The data to be analysed does not have to follow the normal distribution, the dependent variable is binary. The resulting model will have generalized characteristics—it will be applicable for prediction and the efficiency of classification into groups should be better than by chance, i.e., the efficiency of classification should be higher than 50%.

## 4. Results

### 4.1. Support Vector Machines

The defined inputs were used for calculation of a SVM model in C ++ code. The basic parameters are: 22 input continuous variables, 1 output categorical variable, classification type 2, Sigmoid function. 1162 vectors were created for active companies and 1161 vectors for companies in liquidation. The relevance of the model is examined in more detail in Table 3.

**Table 3.** SVM model prediction status.

| | Status—Active Company | Status—In liquidation | Status—All |
|---|---|---|---|
| Total | 4606 | 1582 | 6188 |
| Correct | 4578 | 130 | 4708 |
| Incorrect | 28 | 1452 | 1480 |
| Correct (%) | 99.39 | 8.22 | 76.08 |
| Incorrect (%) | 0.61 | 91.78 | 23.92 |

Source: own construction.

The accuracy of classifications, or predictions is more than 76%. This is certainly positive in terms of the model success. However, remember that this percentage consists of more than 99% of correct predictions of active companies and only above 8% of predictions of the companies in liquidation. Therefore, the model is not fully applicable in practice.

*4.2. Artificial Neural Networks*

10,000 artificial neural structures were calculated of which 5 with the best characteristics were retained (see Table 4).

**Table 4.** Retained neural networks.

| Statistics | 1 | 2 | 3 | 4 | 5 |
|---|---|---|---|---|---|
| Network name | MLP 22-6-2 | MLP 22-9-2 | MLP 22-12-2 | MLP 22-8-2 | MLP 22-12-2 |
| Training performance | 81.46353 | 83.01016 | 82.2253 | 82.40997 | 83.05633 |
| Testing performance | 80.38793 | 81.89655 | 81.03448 | 81.25 | 81.14224 |
| Validation performance | 81.35776 | 82.65086 | 83.40517 | 82.65086 | 83.40517 |
| Training algorithm | BFGS 170 | BFGS 332 | BFGS 56 | BFGS 110 | BFGS 220 |
| Error function | Entropy | Entropy | SOS | Entropy | Entropy |
| Hidden activation func. | Tanh | Tanh | Identity | Logistic | Tanh |
| Output activation func. | Softmax | Softmax | Logistic | Softmax | Softmax |

Source: own construction.

The best characteristics of generated neural structures are exclusively shown by MLP networks. NNs have 22 neurons in the input layer (based on 22 input continuous variables), 6 to 12 neurons in the hidden layer and 2 neurons in the output layer (based on one output categorical variable that can take two values). Entropy was the error function in four cases, the sum of squares in one. The identity, logistic and hyperbolic tangent functions were used to activate the hidden layer of neurons. The logistic and Softmax functions were used to activate the output layer of neurons. The performance of individual networks is always above 81% in the train data set and above 80% in the test data set and above 81% in the validation set. Thus, the performance seems very high. Table 5 shows the performance decomposition.

**Table 5.** Predictions of artificial neural networks.

| Network | Statistics | Status—Active Company | Status—In liquidation | Status—All |
|---|---|---|---|---|
| 1.MLP 22-6-2 | Total | 4606 | 1582 | 6188 |
| | Correct | 4226 | 804 | 5030 |
| | Incorrect | 380 | 778 | 1158 |
| | Correct (%) | 91.75 | 50.82 | 81.29 |
| | Incorrect (%) | 8.25 | 49.18 | 18.71 |
| 2.MLP 22-9-2 | Total | 4606 | 1582 | 6188 |
| | Correct | 4234 | 889 | 5123 |
| | Incorrect | 372 | 693 | 1065 |
| | Correct (%) | 91.92 | 56.20 | 82.79 |
| | Incorrect (%) | 8.08 | 43.81 | 17.21 |
| 3.MLP 22-12-2 | Total | 4606 | 1582 | 6188 |
| | Correct | 4315 | 773 | 5088 |
| | Incorrect | 291 | 809 | 1100 |
| | Correct (%) | 93.68 | 48.86 | 82.22 |
| | Incorrect (%) | 6.32 | 51.14 | 17.78 |
| 4.MLP 22-8-2 | Total | 4606 | 1582 | 6188 |
| | Correct | 4320 | 771 | 5091 |
| | Incorrect | 286 | 811 | 1097 |
| | Correct (%) | 93.79 | 48.74 | 82.27 |
| | Incorrect (%) | 6.21 | 51.26 | 17.73 |
| 5.MLP 22-12-2 | Total | 4606 | 1582 | 6188 |
| | Correct | 4252 | 873 | 5125 |
| | Incorrect | 354 | 709 | 1063 |
| | Correct (%) | 92.31 | 55.18 | 82.82 |
| | Incorrect (%) | 7.69 | 44.82 | 17.18 |

Source: own construction.

Ideally, we are looking for a neural structure which shows the highest number of correctly classified cases. However, it is very important for NN to be able to predict (classify) both active companies (i.e., businesses capable of surviving a potential crunch) and companies in liquidation (i.e., businesses in bankruptcy). In this respect, 2.MLP 22-9-2 and 5.MLP 22-12-2 networks appear to be the most successful. There is a minimum difference between them. But a higher number of correct predictions of bankruptcy for 2.MLP 22-9-2 network is more advantageous. The dominance of both networks is illustrated by the chart in Figure 1.

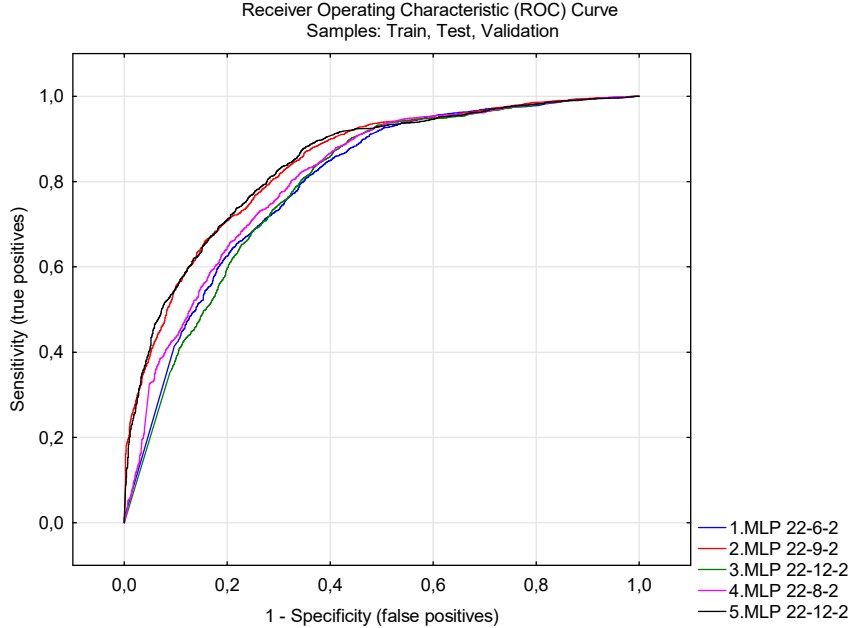

**Figure 1.** Threshold operating characteristics of neural network classification. Source: own construction.

Ideally, the characteristics are close to (0,1). The 2.MLP 22-9-2 and 5.MLP 22-12-2 networks are closest to this point.

### 4.3. SVM/NN Comparison

It is obvious from the results that the SVM model has a quite high level of reliability. However, the structure of correct classifications, i.e., 99% of correct predictions of active companies and only above 8% of predictions of companies in liquidation, makes this model inapplicable.

On the contrary, five NN models were retained by applying the methodology for creating NN. In all cases, those are MLPs that are applicable in practice. There are minimum differences between networks. Still we can identify the best neural network which is NN 2.MLP 22-9-2 without any doubt: very closely followed by NN 5.MLP 22-12-2. There is just a minimum difference between them.

This answers our research question. In this case, the answer is very simple. Artificial neural networks are much more accurate than SVM in predicting possible bankruptcy. Unlike SVM all retained NNs are well applicable in practice.

It is a bankruptcy model. We thus define a tool to identify the companies unlikely to survive a possible financial distress. In particular, we examine the ability of the tool to identify a company that can be expected to face financial distress in the future. The SVM model showed a great ability to predict the second opposite situation at first glance, that is, the ability of the company to survive a possible financial distress. In this case, the prediction of the model is correct in 99.39% of cases. However, the ability to predict bankruptcy is at the 8.22% level. In general, the SVM model predicts the future development of the company with 76.08% accuracy, which could be considered a good result. However, the problem is that the model would achieve the same or almost the same predictive power

even if it did not predict any company that is going to bankrupt. In fact, the SVM method did not meet the requirements, although it shows a rather interesting result. The SVM model is thus nonapplicable.

As the confusion matrix in Table 5 indicates, artificial neural networks show higher prediction power—nearly up to 83%, but what is even more important, they have greater ability to predict companies that are going to bankrupt. Taking into account the most successful neural structure, 2.MLP 22-9-2, its accuracy is 82.79%. It is able to predict correctly 91.92% of companies that are able to survive a potential financial distress, and 56.2% of companies that are going to bankrupt. The prediction is thus applicable in practice.

Now the task is to find a generally acceptable model able to predict a potential financial distress. The Altman Z-Score (Altman 1968, 2000, 2003; Altman and Hotchkiss 2006) and many other models (Neumaierová and Neumaier 2005, 2008; Taffler and Tisshaw 1977; Taffler 1983; Kralicek 1993; Pollak 2003) were based on the data that are not relevant for the current corporate environment (small data volume, data more than 50 years old, etc.). Although the Altman Z-Score is still being used, corporate practice is well aware of their weaknesses. The paper aimed to find an alternative that respect the time lag and which would be easily applicable and showing an appropriate level of accuracy. Very often, it is about being able to detect a potential risk associated with a particular company. Subsequently, we would be able to analyse such a company in more detail, assessing whether the risk is real or not.

This requirement is definitely met by the generated neural networks, in particular 2.MLP 22-9-2. It is based on the current data in the environment where the resulting model of neural networks will be applied. As stated above, it is the first indication of possible problems used as an impulse for a more detailed analysis. The resulting model is interesting from another aspect. Despite its easy applicability, the artificial neural network assesses the future development on the basis of 22 variables characterizing the amount of company assets, structure of its financing, payment history of the company and the customers, cost structure, and the ability to generate sales (as a quantified output of core business). The individual indicators are described in Data and Methods.

Since 2000, many authors have tried to predict company bankruptcy using the models of neural networks. As an example, we can mention Becerra et al. (2002), who analysed the use of linear models and the models of neural networks for the classification of financial distress. Their calculation included 60 British companies from the period between 1997 and 2000. Zheng and Jiang (2007) used the data of Chinese listed companies between 2003 and 2005. All similarly created models are rather outdated, as they use the data that were up to date before the world financial crisis. This paper shows an up-to-date and simple model (most existing studies create relatively complex hybrid models—e.g., Xu et al. 2019), which can be gradually updated using new data, and thus even become more accurate (due to neural networks learning).

## 5. Discussion and Conclusions

Bankruptcy prediction is always a topical issue. This is due to very complicated business relationships between entrepreneurs and competition in the current business environment. It is characterized by instability, perhaps even turbulence. All the more important is to find a low-input tool that can evaluate future development of any company in the market.

The aim of this paper was to develop bankruptcy prediction models and evaluate the results obtained from classification methods, namely Support Vector Machines and artificial neural networks (multilayer perceptron artificial neural networks—MLP and radial basis function artificial neural networks—RBF).

In total, six models were created: 1 SVM, 5 NN. Consequently, a comparison was made between them. NN 2.MLP 22-9-2 appears to be the most successful model that is applicable in practice (NN code C++ forms). The financial variables with the highest bankruptcy predictive power are presented in Table 6.

**Table 6.** Sensitivity analysis.

| Variables | 1.MLP 22-6-2 | 2.MLP 22-9-2 | 3.MLP 22-12-2 | 4.MLP 22-8-2 | 5.MLP 22-12-2 | Average |
|---|---|---|---|---|---|---|
| OHANIM | 1.307736 | 8.298830 | 1.623772 | 1.197549 | 1.143286 | 2.714235 |
| PRIDHODN | 1.302244 | 4.395480 | 1.584667 | 2.339157 | 2.748509 | 2.474011 |
| VS | 1.319040 | 2.663396 | 1.602347 | 3.742576 | 2.887139 | 2.442900 |
| HVML | 1.269237 | 2.125292 | 1.520003 | 1.517179 | 3.173511 | 1.921044 |
| MZDN | 1.294799 | 2.563237 | 1.561902 | 1.494294 | 2.231695 | 1.829185 |
| OBEZNAA | 1.274424 | 2.737114 | 1.627830 | 1.209418 | 2.123338 | 1.794425 |
| SPMAAEN | 1.324918 | 2.146751 | 1.295759 | 1.266527 | 2.599045 | 1.726600 |
| STALAA | 1.173915 | 2.153740 | 1.231161 | 1.038480 | 2.572654 | 1.633990 |
| Z | 1.289484 | 2.095527 | 1.494067 | 1.115624 | 1.585507 | 1.516042 |
| V | 1.315965 | 2.092471 | 1.608308 | 1.113233 | 1.146686 | 1.455333 |
| FTZZ | 1.278720 | 1.379155 | 1.539660 | 1.668709 | 1.389535 | 1.451156 |
| CIZIZDROJE | 1.527338 | 1.269488 | 1.853573 | 1.073045 | 1.487208 | 1.442131 |
| SLUZBY | 1.422673 | 1.335220 | 2.002430 | 1.127664 | 1.204212 | 1.418440 |
| FM | 1.076257 | 1.418978 | 1.601751 | 1.185785 | 1.525021 | 1.361559 |
| HVUO | 1.298108 | 1.459786 | 1.454034 | 1.219113 | 1.350517 | 1.356312 |
| KZ | 1.258229 | 1.441923 | 1.204971 | 1.326370 | 1.334837 | 1.313266 |
| HIM | 1.095701 | 1.904764 | 1.004551 | 1.328013 | 1.228624 | 1.312330 |
| VLASTNIJM | 1.288897 | 1.338126 | 1.526678 | 1.225678 | 1.160983 | 1.308072 |
| KP | 1.280438 | 1.581155 | 1.337826 | 1.034392 | 1.196151 | 1.285992 |
| NNSOCZAB | 1.016164 | 1.991251 | 1.058640 | 1.060163 | 1.284383 | 1.282120 |
| AKTIVACELK | 1.274310 | 1.452314 | 1.154554 | 1.014583 | 1.388433 | 1.256839 |
| PASIVACELK | 1.274334 | 1.446461 | 1.154320 | 1.014663 | 1.368253 | 1.251606 |

Source: own construction.

The highest bankruptcy predictive power have "Depreciation of intangible and tangible fixed assets", "Value added" and "Production consumption". All three items are logical for the manufacturing industry.

The existing models (Altman index, Neumaier index and many others) are based on the standard statistical methods. Their deficiencies were identified by Balcaen and Ooghe (2004):

- Dependent variable dichotomy,
- Sampling method,
- Stationarity and data instability,
- Selection of variables,
- Using information from financial statements, and
- Time dimension.

Neural networks can resolve some of the defined problems. It is primarily the time dimension. For all the existing models, the previous development of the company, consequently evaluated as Active or in Liquidation, cannot be taken into account. Neural networks are able to handle large data volumes. Therefore, the values of variables of selection do not need to be restricted. It may appear that the dataset will be a limit when application for another period and different market (especially when used abroad). However, it is not the case, as we identified a structure with a relatively strong prediction power. Although it was trained and subsequently validated twice on a selected sample, the neural network can be quickly adapted to the specificities of a different market. Artificial neural network can adapt to a new environment by retraining it on a dataset sample of a given market. Due its ability to meet the requirement for changing the setting of its internal parameters, neural network can thus be considered flexible and widely applicable.

The future focus should to collect data other than information from financial statements. It will also be necessary to define the company status other than just Active or in Liquidation. However, the data problem may not be resolved.

**Author Contributions:** Conceptualization, J.H. and J.V.; methodology, J.H. and J.V.; software, P.S.; validation, J.H., J.V. and P.S.; formal analysis, J.H.; investigation, J.V. and P.S.; resources, J.H.; data curation, J.H.; writing—original draft preparation, J.H. and J.V.; writing—review and editing, P.S.; visualization, J.H. and J.V.; supervision, P.S. All authors have read and agreed to the published version of the manuscript.

**Funding:** This research received no external funding.

**Conflicts of Interest:** The authors declare no conflict of interest. The funders had no role in the design of the study; in the collection, analyses, or interpretation of data; in the writing of the manuscript, or in the decision to publish the results.

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
