# Peer review of "Support Vector Machine Methods and Artificial Neural Networks Used for the Development of Bankruptcy Prediction Models and their Comparison"

_jrfm, doi:10.3390/jrfm13030060_

Round 1

Reviewer 1 Report

This paper uses Support Vector Machine and artificial neural networks methods to predict bankruptcy in Czech Republic. The results show that artificial neural networks are much more accurate than SVM in predicting possible bankruptcy.

I find the paper well developed and written. The two approaches are applied and contrasted. I suggest including additional macroeconomic indicators in the predictive model to account for business cycles or institutional changes. Moreover, how do the models perform in the out-of-sample forecasting?  

For the comparison of the two kinds of models, you may want to provide more explanations for the strengths and weakness, and show why NM would outperform SVM.

Author Response

Dear reviewer,

Please find attached all necessary information.

Thank you for your feedback.

With kind regards

Jakub Horak

Reviewer 2 Report

See file attached

Author Response

(The authors gave the same response as above.)
